# Identification of sSIGLEC5 and sLAG3 as New Relapse Predictors in Lung Cancer

**DOI:** 10.3390/biomedicines10051047

**Published:** 2022-04-30

**Authors:** Karla Montalbán-Hernández, José Carlos Casalvilla-Dueñas, Patricia Cruz-Castellanos, Laura Gutierrez-Sainz, Roberto Lozano-Rodríguez, José Avendaño-Ortiz, Carlos del Fresno, Javier de Castro-Carpeño, Eduardo López-Collazo

**Affiliations:** 1The Innate Immune Response Group, IdiPAZ, La Paz University Hospital, 28046 Madrid, Spain; karlamarina.hernandez@idipaz.es (K.M.-H.); josecarlos.casalvilla.duenas@idipaz.es (J.C.C.-D.); roberto.lozano.rodriguez@idipaz.es (R.L.-R.); jose.avendano@salud.madrid.org (J.A.-O.); carlos.delfresno.sanchez@idipaz.es (C.d.F.); 2Tumour Immunology Lab, IdiPAZ, La Paz University Hospital, 28046 Madrid, Spain; 3Medical Oncology Department, La Paz University Hospital, 28046 Madrid, Spain; patricia.cruz@salud.madrid.org (P.C.-C.); lgutierrezs@salud.madrid.org (L.G.-S.); 4Centre for Biomedical Research, Network of Respiratory Diseases (CIBERES), 28029 Madrid, Spain; 5Immunomodulation Lab, IdiPAZ, La Paz University Hospital, 28046 Madrid, Spain

**Keywords:** lung cancer, sSiglec5, sLAG3, relapse, predictive

## Abstract

Lung cancer (LC) continues to be the leading cause of cancer-related deaths in both men and women worldwide. After complete tumour resection, around half of the patients suffer from disease relapse, emphasising the critical need for robust relapse predictors in this disease. In search of such biomarkers, 83 patients with non-microcytic lung cancer and 67 healthy volunteers were studied. Pre-operative levels of sSIGLEC5 along with other soluble immune-checkpoints were measured and correlated with their clinical outcome. Soluble SIGLEC5 (sSIGLEC5) levels were higher in plasma from patients with LC compared with healthy volunteers. Looking into those patients who suffered relapse, sSIGLEC5 and sLAG3 were found to be strong relapse predictors. Following a binary logistic regression model, a sSIGLEC5 + sLAG3 score was established for disease relapse prediction (area under the curve 0.8803, 95% confidence intervals 0.7955–0.9652, cut-off > 2.782) in these patients. Based on score cut-off, a Kaplan–Meier analysis showed that patients with high sSIGLEC5 + sLAG3 score had significantly shorter relapse-free survival (*p* ≤ 0.0001) than those with low sSIGLEC5 + sLAG3 score.Our study suggests that pre-operative sSIGLEC5 + sLAG3 score is a robust relapse predictor in LC patients.

## 1. Introduction

Non-small cell lung cancer (NSCLC), which composes 85% of lung cancers, continues to be the leading cause of cancer-related deaths in both men and women worldwide since decades [1]. Furthermore, lung cancer (LC) shows a five-year survival rate below 15%, one of the lowest within the different types of cancers [2,3]. Symptom onset usually occurs once damage is too extensive, meaning that most cases are diagnosed at advanced stages [4]. Moreover, the presence of metastasis at diagnosis is one of the major reasons for treatment failure [5]. Low dose spinal computed tomography has increased survival and resectability rates due to its ability of providing an effective early diagnosis; however, there is a lack of annual screening [6]. Nonetheless, when these patients are diagnosed early, they can be considered for curative surgery. In fact, among those diagnosed individuals, stage I LC patients usually undergo surgery as first-line of therapy. However, around 30% of these stage I patients later develop disease relapse [7], even after complete tumour resection [8]. Altogether, around half of LC patients suffer from disease relapse, even after complete tumour resection [8].

Therefore, there is a clear need to generate robust predictive models for survival and disease relapse in LC patients. Early identification of patients at high risk of relapse after surgery is crucial for a correct monitoring and treatment regime to improve overall survival. Several models have already been developed, most of them involving a combination of clinical data and molecular markers [9,10]. Gene expression analysis has been used for determination of prognostic signatures, but none have shown to be potent enough for clinical application [11].

LC has joined kidney cancer and melanoma as one of the best indicated tumours for immunotherapy [12]. Immunotherapy against the programmed death ligand-1 (PD-L1) and programmed death-1 (PD-1) axis has significantly increased survival in lung cancer patients [13]. Blockade of this axis with FDA-approved immune-checkpoint (IC) inhibitors therapy, such as nivolumab and pembrolizumab, releases T lymphocytes to recognise and attack tumour cells. In this context, PD-L1 expression in tumours emerged as the first biomarker with predictive value to determine those patients who could benefit from IC therapy [14,15]. Although increased levels of soluble PD-L1 (sPD-L1) have been found in IC therapy responder patients, PD-L1 expression heterogeneity within tumours has compromised its predictive value [16,17]. Looking for alternative markers, lymphocyte activation gene-3 (LAG3) and T-cell immunoglobulin and mucin-domain containing-3 (Tim-3) also form part of the most relevant IC in LC treatment. However, the prognostic strength of their quantification in LC remains controversial.

In this regard, amongst immune-regulatory molecules, sialic acid-binding immunoglobulin-type lectins (SIGLECs) are expressed on innate immune cells and can modulate immune responses in cancer, making them highly interesting IC ligands [18]. These lectins, in a similar way to widely established IC such as PD-L1 and PD-1, possess an amino-terminal V-set immunoglobulin domain that recognises sialic acids [19]. Mechanistically, hypersialylation of cancer cells favours the interaction with inhibitory SIGLECs which supress the immune response, and this binding is associated with tumour progression [20,21]. Among this receptor family, SIGLEC5 and its soluble form, soluble SIGLEC5 (sSIGLEC5), have been shown to bind the highly sialylated P-selectin glycoprotein ligand-1 (PSGL-1) molecule present on T lymphocytes, promoting tumour progression [22,23].

The use of liquid biopsy to establish survival and relapse biomarkers is rapidly growing. However, it is hard to find individual biomarkers with high prediction value, which is why combination models are widely used [24,25]. Therefore, a critical need for the identification of predictive biomarkers exists in clinical practice. The function of soluble receptors and ligands has been poorly studied in cancer, considering the roles these play in immune regulation. In LC patients, sPD1 levels have been associated to erlotinib response and overall survival [26]. Levels of well-known soluble IC have been associated to disease prognosis; such as elevated soluble CD86 (sCD86) in myeloma; however, in the case of LC, no associations have been made to the best of our knowledge [27]. Of note, sSIGLEC5 has been recently demonstrated to be a prognosis marker and exitus predictor in colorectal cancer [28].

Herein, we hypothesised sSIGLEC5 might act as a disease relapse predictor in LC. First, we found pre-operative sSIGLEC5 concentration to be higher in plasma samples from LC patients compared to healthy volunteers. Importantly, among patients, sSIGLEC5 levels were significantly higher in those who suffered relapse. Searching for further IC associated with relapse in our LC cohort, sLAG3 appeared to be also strongly associated with disease recurrence. Ultimately, the combination of both sSIGLEC5 and sLAG3 showed the strongest predictive capacity of disease relapse. These findings support the combination of sSIGLEC5 and sLAG3 as clinical markers to define high-risk of relapse LC patients, and therefore, a novel and robust tool to identify patients worth of a close monitoring.

## 2. Materials and Methods

### 2.1. Patient and Healthy Volunteer Recruitment

Patients diagnosed with LC (*n* = 83) were enrolled in this study prior to complete surgical resection at La Paz University Hospital (Madrid, Spain). Exclusion criteria included metastasis at diagnosis (stage IV) and neoadjuvant treatment. Patients were followed-up for a median time of 1 year and were classified according to their disease stage (Table 1). Blood samples were taken 24 h prior to the surgery and plasma was isolated following standardised procedures [29]. As controls, healthy volunteers (HV) (*n* = 67) were recruited from the Blood Donor Services of La Paz University Hospital (Table 1).

### 2.2. Ethics Approval

Consent forms were signed by all patients and HV involved in the study. The ethical guidelines of the 1975 Declaration of Helsinki were followed to ensure data were treated according to the recommended criteria of confidentiality. The local Ethics Committee (La Paz University Hospital, Madrid, PI-3521) approved this study.

### 2.3. ELISA Assay

A commercially available enzyme-linked immunosorbent assay (ELISA) kit (Sigma-Aldrich, Sant Luis, MO, USA) was used to determine sSIGLEC5 concentration in plasma samples from LC patients and HVs. The manufacturer’s instructions were followed to determine these concentrations.

### 2.4. Soluble Immune Checkpoint Measurement

Soluble ICs (sB7.2, sCD25, sLAG3, sPD-1, sPD-L1, sTim3 and s4-1BB) were measured in plasma samples from patients using the LegendPlex Custom Human Immune Checkpoint Panel, following the manufacturer’s instructions (Biolegend, San Diego, CA, USA). Briefly, plasma samples were incubated with premixed capture antibody-coated beads, washed, incubated with detection antibodies and streptavidin with phycoerythrin conjugate, acquired using a FACSCalibur flow cytometer (BD) and analysed with Biolegend v8.0 software (Biolegend).

### 2.5. Statistical Analysis

Continuous variables were analysed by either parametric or nonparametric statistical tests after analysing their Gaussian distribution by the Shapiro–Wilk test. Outliers were removed from their corresponding variable using ROUTs test by GraphPad prism 9.3.1. Categorical parameters were analysed by chi-squared tests. To analyse the differences between disease stages, a one-way ANOVA with Tukey’s post-hoc test was used. The correlation between variables were analysed with Pearson’s r correlations and linear regression models. In order to compare relapse-free survival times between groups, both the receiver operating characteristic (ROC); with the Wilson/Brown test, and Kaplan–Meier survival curves; with both the Cox Mantel log-rank and Gehan–Breslow Wilcoxon tests, were used (GraphPad Prism 9). The logistic regression model was performed with SPSS version 23 (IBM) software. The score of the regression model (Appendix A) can be calculated with the following formula: Score = (sLAG3 × 0.001606) + (sSIGLEC5 × 0.003875). Additionally, Microsoft Office Excel and GraphPad Prism 9 were used to calculate the Youden index and the area under the curve (AUC) with 95% confidence intervals (Cis) respectively. *p* values of *, *p* < 0.05; **, *p* < 0.01; ***, *p* < 0.001; ****, *p* < 0.0001 were considered significant. *p* values of *, *p* < 0.05; **, *p* < 0.01; ***, *p* < 0.001; ****, *p* < 0.0001 were considered significant.

## 3. Results

### 3.1. Patient Characteristics

From 28 November 2019, to 6 August 2021, the Thoracic Surgery Service of La Paz University Hospital in Madrid, Spain consecutively recruited a total of 83 patients with LC. Blood was collected 24 h prior to complete surgical resection. Patients were followed-up until 23 November 2021, with continuous check-ups, and were classified into three groups according to their disease stage: I, II or III. Table 1 summarises the patients’ characteristics, including: tumour histology, metastasis, comorbidities, patient outcome and disease relapse. As a control group, HVs (*n* = 67) were recruited by the Blood Donor Services of La Paz University Hospital and were also assessed in this cohort.

**Table 1 biomedicines-10-01047-t001:** Patient and healthy volunteers characteristics ^1^.

Characteristic	Healthy Volunteers *n* = 67	All Patients *n* = 83	Stage I *n* = 55	Stage II *n* = 16	Stage III *n* = 12	*p*-Value
Gender						0.9680
Male	29 (43)	54 (65)	36 (67)	10 (19)	8 (14)	
Female	38 (56)	29 (35)	19 (66)	6 (21)	4 (13)	
Age						0.9695
Median Range	59 (50–75)	67 (38–86)	67 (38–86)	67 (61–77)	65 (58–83)	
Relapse		13 (15)	6 (47)	5 (38)	2 (15)	0.1428
Perineural Invasion		3 (3)	1 (33.3)	1 (33.3)	1 (33.3)	0.4504
Linfovascular Invasion		11 (13)	2 (18)	5 (46)	4 (36)	0.0014
Tumour Histology						
Adenocarcinoma		58 (69)	43 (74)	8 (13)	7 (12)	0.0619
Epidermoid		21 (25)	12 (57)	6 (29)	3 (14)	0.4463
Large Cell Neuroendrocrine Carcinoma		2 (2)	0 (0)	2 (100)	0 (0)	0.0137
Others		2 (2)	0 (0)	0 (0)	2 (100)	0.0023
Adjuvant Treatment						
Chemotherapy		18 (22)	0 (0)	8 (44)	10 (56)	0.0001
Radiotherapy		7 (8)	2 (29)	1 (14)	4 (57)	0.0034
Immunotherapy		1	0 (0)	0 (0)	1 (100)	0.0501
Comorbidities						
Smoker		67 (80)	47 (70)	13 (20)	7 (10)	0.0973
Exitus		5 (6)	2 (40)	2 (40)	1 (20)	0.3961

^1^ Data are presented as *n* (% or range). *p* values show significant differences between patient groups.

### 3.2. sSIGLEC5 Is a Relapse Predictor in Lung Cancer Patients

The concentrations of sSIGLEC5 were determined in pre-operative plasma samples from all participants. Figure 1A shows significantly higher sSIGLEC5 levels in patients with LC (*n* = 83) compared with HVs (*n* = 67) (*p* ≤ 0.0001). This result resembled the behaviour of sSIGLEC5 in colorectal cancer patients [28]. However, no significant correlations were observed between disease stages or exitus in LC patients (Appendix A). Nevertheless, sSIGLEC5 levels were significantly higher in patients who suffered relapse (Figure 1B). Given that pre-operative sSIGLEC5 levels were higher in relapse, we explored whether sSIGLEC5 plasma levels could serve as relapse predictors in LC. To discriminate between patients who suffered or not from disease relapse, a ROC analysis of the plasma sSIGLEC5 levels was performed (Figure 1C, AUC = 0.727; 95% CI 0.5930–0.8620; *p* = 0.0095). The optimal cut-off value, estimated by the Youden index, was 382 ng/mL and exhibited a high sensitivity (0.92; 95% CI 0.666–0.996), and a specificity of 0.51 (95% CI 0.399–0.627). 

The obtained pre-operative Youden-cut off value allowed LC patient classification into sSIGLEC5-High and sSIGLEC5-Low groups. The sSIGLEC5-High group had a significantly higher relapse probability than the sSIGLEC5-Low group (Figure 1D, χ^2^ = 8.489, *p* = 0.0036). In order to compare the differences in relapse free survival times between both groups, Kaplan–Meier analyses were carried out. The sSIGLEC5-High group showed a significantly shorter relapse-free survival than the sSIGLEC5-Low group (Figure 1E). Both the Gehan–Breslow–Wilcoxon and log-rank (Mantel-cox) tests showed statistical significance, indicating the potent role of pre-operative sSIGLEC5 concentration as a relapse predictor in LC patients.

### 3.3. sLAG3 Is Increased in Relapse Lung Cancer Patients

Considering the capacity of sSIGLEC5 to discriminate relapse in LC patients, we wondered whether the presence of other soluble IC in pre-operative plasma samples could act as relapse predictors. Thus, the concentrations of sPD-1, sPD-L1, sB7.2, sCD25, sTim3, s4-1BB and sLAG3 were also determined in these samples. No significant differences in sPD-1, sPD-L1, sB7.2, sCD25, sTim3 or s4-1BB levels between patients who suffered or not from relapse were found (Figure 2A–F). Interestingly, sLAG3 levels were significantly higher in patients who suffered relapse (Figure 2G).

### 3.4. sLAG3 Can Act as a Relapse Predictor in Lung Cancer Patients

Given that pre-operative plasma levels of sLAG3 was found elevated in patients who suffered relapse (Figure 2), we moved on to study whether pre-operative sLAG3 plasma levels could also serve as a relapse predictor in lung cancer. ROC analysis was performed in order to obtain cut-off values to discriminate between patients who suffered or not from relapse. To discriminate between patients who suffered or not from disease relapse, a ROC analysis of the plasma sLAG3 levels was performed (Figure 3A, AUC = 0.867; 95% CI 0.770–0.965; *p* = < 0.0001). The optimal cut-off value, estimated by the Youden index, was 722.5 pg/mL and exhibited a high sensitivity (0.84; 95% CI 0.577–0.972), and a specificity of 0.70 (95% CI 0.587–0.807).

Based on the Youden index analyses, patients were classified into sLAG3 -Low or -High groups. Subsequently, relapse-free survival times between both groups were compared using Kaplan–Meier analyses. Figure 3C shows the sLAG3-High group had a significantly shorter relapse-free survival time than the sLAG3-Low group. Both the Gehan-Breslow–Wilcoxon and log-rank (Mantel-cox) tests showed statistical significance. Altogether, the data indicated the potential of sLAG3 as a relapse predictor in LC patients. Therefore, similarly as observed for sSIGLEC5, plasma levels of sLAG3 are good relapse predictors in LC patients. Contingency table analyses showed a significantly higher relapse probability in the sLAG3-High group compared with the sLAG3-Low group (Figure 3, χ^2^ = 10.00, *p* = 0.0016).

### 3.5. Combination of sSIGLEC5 and sLAG3 Increases Robustness of the Predictive Model

After establishing sSIGLEC5 and sLAG3 as relapse predictors in lung cancer, we moved on to determine whether levels of both correlated and whether their combination could provide a more robust relapse prediction model. Figure 4A shows sSIGLEC5 and sLAG3 levels have a significant positive correlation. Next, the score obtained for the combination of sSIGLEC5 and sLAG3, following a binary logistic regression model (Appendix A), was significantly higher in patients who suffered relapse (Figure 4B). ROC curve analysis was performed to determine a cut-off score for the combination of sSIGLEC5 and sLAG3 (Figure 4C, AUC = 0.8803; 95% CI 0.7955–0.9652; *p* = < 0.0001). The optimal cut-off score, estimated by the Youden index, was 2.782 and exhibited a high sensitivity (1; 95% CI 0.771–1.000) and specificity (0.66; 95% CI 0.543–0.770). The latter analysis confirmed the combination of sSIGLEC5 and sLAG3 and produced a more potent predictive model compared to both sSIGLEC5 and sSLAG3 on their own. Contingency table analyses based on the Youden index showed a strong higher relapse probability in the sSIGLEC5 + sLAG3-High group compared with the sSIGLEC5 + sLAG3-Low group (Figure 4D, χ^2^ = 20.01, *p* < 0.0001). Patients were then classified into sSIGLEC5 + sLAG3-High and sSIGLEC5 + sLAG3-Low groups according to their Youden cut-off score. Kaplan–Meier analysis showed that patients belonging to the sSIGLEC5 + sLAG3-High group had a significantly shorter relapse-free survival than those in the sSIGLEC5 + sLAG3-Low group (Figure 4E). All these data together indicate that the combination of plasma sSIGLEC5 and sLAG3 levels represents a robust predictor for LC relapse prediction at the pre-operative moment.

## 4. Discussion

Lung cancer continues to be the leading cause of death in both men and women worldwide, with a 5-year survival rate below 15% [1,3]. Due to the fact that most cases are diagnosed once the disease is advanced, metastases are common at diagnosis, which worsens treatment response [5]. Immunotherapy has become together with chemotherapy and radiotherapy one of the gold standard treatments, with the use of targeted therapy in specific tumour types [30,31]. In NSCLC, which comprises the majority of LC tumours, surgery is only recommended for early stages of the disease [32]; however, even after surgery, a great number of patients suffer relapse, with a greater risk of more advanced stages. Therefore, there is a need for predictive biomarkers of prognosis, which is further complicated because of disease heterogeneity between patients.

Recent randomised trials have shown how combination of IC blockade therapy together with chemotherapy has improved progression-free survival times from 4.5 to 6.2 months [33]. However, other trials have shown the use of IC blockade therapy in advanced NSCLC patients, even though it doubled progression-free survival, it showed no benefit in overall survival [34]. Standardised treatment in our cohort of patients after surgery involved the use of adjuvant chemotherapy (four cycles of cisplatin) in tumours bigger than 4 cm or with lymph node involvement. In a similar manner to the described trials, survival in these patients only slightly increased around 5% [35]. Despite the rapid advances in LC diagnosis and treatments indicated above, disease relapse continues to be more of a rule than an exception [36].

Nowadays, no biomarkers exist to predict disease relapse in LC patients. Herein, we have reported an incremented concentration of pre-operative sSIGLEC5 in plasma samples from LC patients. Looking into those patients who suffered relapse, sSIGLEC5 and sLAG3 were found to be significantly higher. Moreover, ROC analyses together with the Youden index showed strong AUCs for sSIGLEC5 and sLAG3 as relapse predictors in LC. Taking all the latter into account, a logistic regression model was used to establish a relapse prediction model with the combination of pre-operative sSIGLEC5 and sLAG3 levels. This model increased the robustness of relapse prediction, with the highest AUC. Altogether, our model provides a novel minimally invasive, rapid and robust pre-operative model for disease relapse prediction. Not only that, but sSIGLEC5 + sLAG3 score will aid stratify patients at higher risk of relapse for close monitoring. Considering SIGLEC5 and LAG3 are immunomodulators, it is important to highlight none of the patients included in the study were metastatic at diagnosis (stage IV) or received neoadjuvant treatment. Due to this exclusion criteria, only patients from stages I to IIIA were included in the study. This further reinforces the value of our markers as strong predictors of disease relapse in our LC patient cohort.

The structure of SIGLEC5 exhibits strong similarities with that from well-known IC candidates, and it has recently been described as a patent IC candidate in sepsis [37]. The hypersialylation present in the cancer cells might be recruiting and interacting with sSIGLEC5, helping the tumour cells escape from immune surveillance. Additionally, our group has previously described the role of sSIGLEC5 as a prognosis marker in colorectal cancer [28]. On the other hand, LAG3 has become one of the most promising IC therapies, although its use has not yet been approved for treatment by the Food and Drug Administration (FDA) [38]. Authors have shown interaction between LAG3 and its receptor FGL1 can cause changes in the tumour microenvironment, such as reduction in IL-2 levels, disrupting anti-tumour immunity [39]. Interestingly and likely in line with our results, LAG3 expression levels on tumour infiltrating lymphocytes (TILs) of LC patients has been shown to correlate with early disease relapse and poor prognosis [40]. Moreover, elevated LAG3 expression has also been associated with resistance to PD-1 treatment [41].

Finally, we speculate that pre-operative sSIGLEC5 and sLAG3 levels might not only be predictors of relapse, but also potential biomarkers to stratify patients for treatment selection. In the same way as PD-L1 expression levels are used to determine which patients could benefit from Pembrolizumab, and taking into account SIGLEC5 and LAG3 are IC [37], our model could work in a similar manner. In this regard, further experimentation is necessary to establish both the origin of sSIGLEC5 and LAG3 and to define their mechanism of action in LC. Additionally, considering the lack of liquid biomarkers in LC and the high relapse rate found in this disease, our data support the use of this combinatorial model as a relapse predictor. The ability to indicate high relapse risk patients prior to surgery, with the ease of sSIGLEC5 + sLAG3 quantification in blood samples drawn 24 h prior to the surgery, could easily be implemented into routine clinical practice for close patient monitoring. Altogether, pre-operative sSIGLEC5 + sLAG3 score could serve as a novel biomarker to be included in pre-operative LC patient’s blood tests.

## Figures and Tables

**Figure 1 biomedicines-10-01047-f001:**
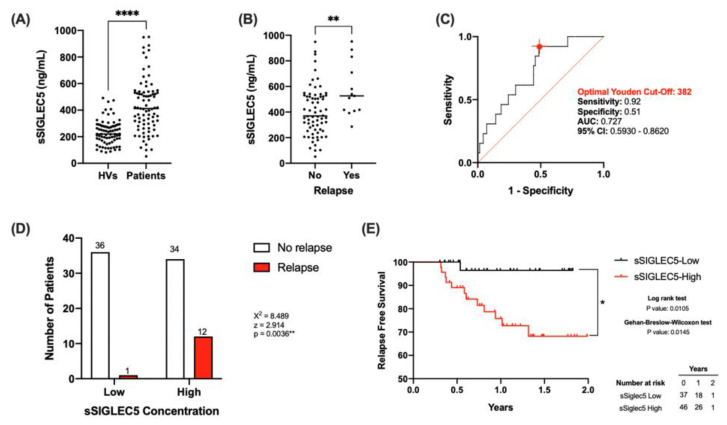
Pre-operative soluble SIGLEC5 (sSIGLEC5) levels act as relapse predictors in lung cancer. (**A**) The soluble SIGLEC5 (sSIGLEC5) levels in plasma from healthy volunteers (HV, *n* = 67) and patients with lung cancer (*n* = 83) quantified by enzyme-linked immunosorbent assay are shown. The difference between groups was analysed by an unpaired Mann–Whitney U test; **** *p* < 0.0001. (**B**) sSIGLEC5 levels in patients who suffered relapse (*n* = 13) vs those who did not (*n* = 70) are shown. The difference between groups was analysed by an unpaired *t* test; ** *p* < 0.01. (**C**) Receiver-operating-characteristic (ROC) curve describing the predictive performance value of plasma sSIGLEC5 for relapse in patients with lung cancer (*n* = 83) (black line; area under the curve [AUC], 0.727 [95% CI 0.593–0.862]) is shown. Optimal cut-off, estimated by the Youden index for plasma sSIGLEC5 concentration, 382 ng/mL (red point shows optimal Youden cut-off specificity and sensitivity values). The ROC curve analysis was performed by Wilson/Brown test, with 95% confidence interval. (**D**) Chart shows number of patients who suffer relapse in the sSIGLEC5-High group (>382 ng/mL) compared with the sSIGLEC5-Low group (<382 ng/mL). Differences between groups were analysed with the chi-squared test, ** *p* = 0.0036. (**E**) Kaplan–Meier curve from surgery date to relapse or end of study date, according to plasma sSIGLEC5 is shown. The differences between relapse free survival rates associated with sSIGLEC5 were calculated by a log-rank (Mantel-Cox; * *p* = 0.0105) test and Gehan–Breslow–Wilcoxon (* *p* = 0.0145) test with 95% confidence interval.

**Figure 2 biomedicines-10-01047-f002:**
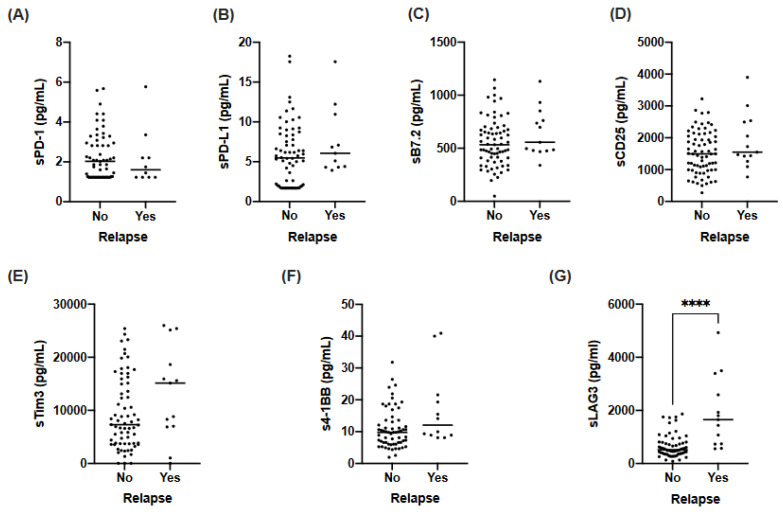
Pre-operative levels of soluble immune-checkpoints (sPD-1, sPD-L1, sB7.2, sCD25, sLAG3, sTim3 and s4-1BB) in lung cancer patients who suffered or not from disease relapse. (**A**) sPD-1 levels in plasma samples from LC patients. (**B**) sPD-L1 levels in plasma samples from LC patients. (**C**) sB7.2 levels in plasma samples from LC patients. (**D**) sCD25 levels in plasma samples from LC patients. (**E**) sTim3 levels in plasma samples from LC patients. (**F**) s4-1BB levels in plasma samples from LC patients. (**G**) sLAG3 levels in plasma samples from LC patients. Differences between patients were analysed by either Mann–Whitney U test; **** *p* < 0.0001 (**A**,**B**,**D**–**G**) or Unpaired *t*-test (**C**).

**Figure 3 biomedicines-10-01047-f003:**
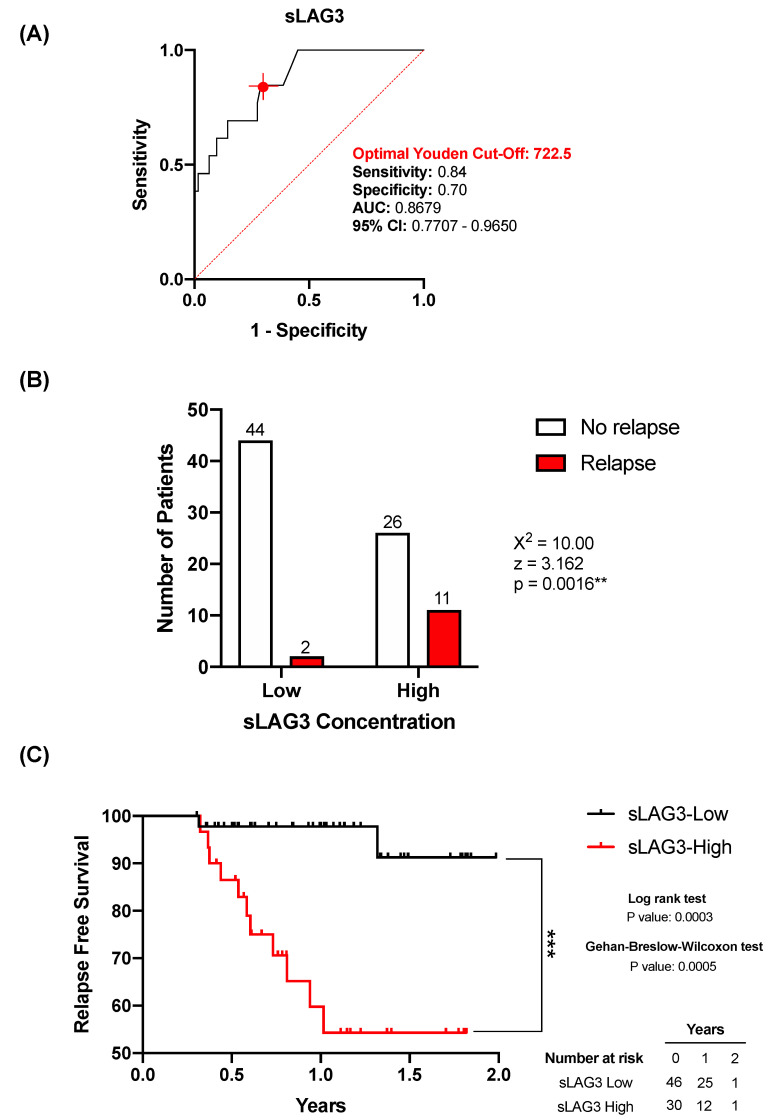
Association of pre-operative levels of sLAG3 and relapse in lung cancer. (**A**) Receiver-operating-characteristic (ROC) curve describing the predictive performance value of plasma sLAG3 for relapse in patients with lung cancer (*n* = 83) (black line; area under the curve [AUC], 0.867 [95% CI 0.770–0.965]) is shown. Optimal cut-off, estimated by the Youden index for plasma sLAG3 concentration, 722.5 pg/mL (red point shows optimal Youden cut-off specificity and sensitivity values). The ROC curve analysis was performed by Wilson/Brown test, with 95% confidence interval. (**B**) Chart shows number of patients who suffer relapse in the sLAG3-High group (>722.5 ng/mL) compared with the sLAG3-Low group (<722.5 ng/mL). Differences between groups were analysed with the chi-squared test, ** *p* = 0.0016. (**C**) Kaplan–Meier curve from surgery date to relapse or end of study date, according to plasma sLAG3 is shown. The differences between relapse free survival rates associated with sSIGLEC5 were calculated by a log-rank (Mantel-Cox; *** *p* = 0.0003) test and Gehan–Breslow–Wilcoxon (*** *p* = 0.0005) test with 95% confidence interval.

**Figure 4 biomedicines-10-01047-f004:**
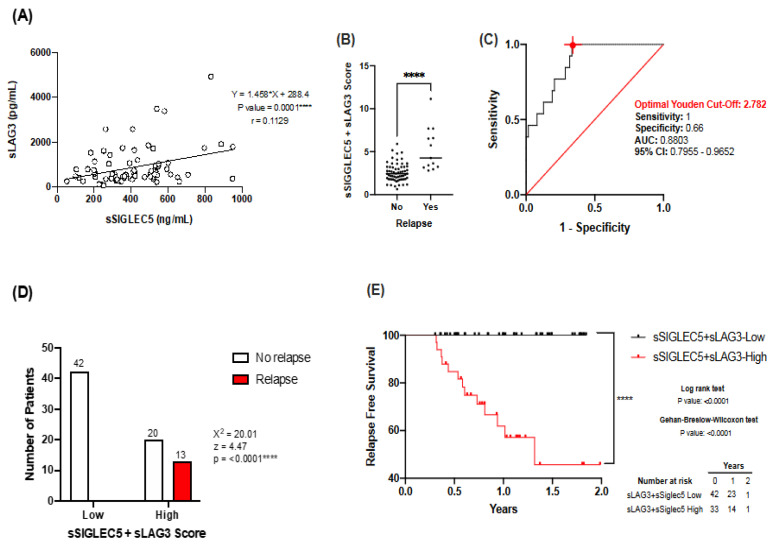
Combination of sSIGLEC5 and sLAG3 generates a strong relapse prediction model. (**A**) Correlation between pre-operative sSIGLEC5 and sLAG3 plasma levels in patients with LC is shown. The correlation between sSIGLEC5 and sLAG3 was analysed by Pearson’s r. Simple correlations of every stage were performed with 95% confidence intervals. (**B**) The sSIGLEC5 + sLAG3 score in plasma from patients who suffered or not from LC relapse are shown. The difference between groups was analysed by an unpaired Mann-Whitney U test, 95% confidence interval; **** *p* < 0.0001. (**C**) The receiver-operating-characteristic (ROC) curve describing the predictive performance value of plasma sSIGLEC5 + sLAG3 for disease relapse in patients with LC (*n* = 75) (black line; area under the curve [AUC], 0.8803 [95% CI 0.795–0.965]) is shown. Optimal cut-off, estimated by the Youden index for plasma sSIGLEC5 + sLAG3 score, 2.782 (red point shows optimal Youden cut-off specificity and sensitivity values) is shown. The ROC curve analysis was performed by Wilson/Brown test, with 95% confidence interval. (**D**) Chart shows number of patients who suffer relapse in the sSIGLEC5 + sLAG3-High group (*n* = 33) compared with the sSIGLEC5 + sLAG3-Low group (*n* = 42). Differences between groups were analysed with the chi-squared test. (**E**) Kaplan–Meier curve from surgery date to relapse or end of study date, according to sSIGLEC5 + sLAG3 score is shown. The differences between relapse free survival rates were calculated by a log-rank (Mantel-Cox; **** *p* = < 0.0001) test with 95% confidence interval. The Gehan–Breslow–Wilcoxon test also showed statistically significant levels (**** *p* = < 0.0001).

## Data Availability

Data are contained within the article and the Appendix A.

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
