# Peer review of "Identification of sSIGLEC5 and sLAG3 as New Relapse Predictors in Lung Cancer"

_biomedicines, 2022, doi:10.3390/biomedicines10051047_

Round 1

Reviewer 1 Report

The manuscript entitled "Identification of sSIGLEC5 and sLAG3 as new relapse predictors in lung cancer" highlighted that pre-operative sSIGLEC5+sLAG3 score is a robust relapse predictor in LC patients.

  • The Authors should provide the expand forms for all acronyms, including gene acronyms, through the text when they first appear.
  • Gene acronyms should be written in italics.

Author Response

Dear reviewer, the expanded form of all acronyms has been included throughout the text when they first appear and can be seen in the corrected manuscript. No gene acronyms were used for this work.

All changes in the corrected manuscript have been done with the “track changes” in yellow to allow easy visualization. 

Reviewer 2 Report

I would like to congratulate the authors for their interesting and informative paper.

This is a study investigating the prognostic significance of sSIGLEC5 and sLAG3 plasma levels in patients with non-small cell lung cancer. The study included 83 patients with lung cancer who underwent upfront surgical resection and 67 healthy individuals. The authors found that patients with high sSIGLEC5+sLAG3 score had significantly shorter relapse-free survival than those with low sSIGLEC5+sLAG3 score.

This is a well written paper. Here, I have made a few suggestions that, in my opinion, could help improve the overall quality of the manuscript.

The authors may consider reporting if all patients underwent complete surgical resection (as opposed to microscopic or macroscopic residual disease).

The authors may consider reporting adjuvant treatments administered to the patients.

The authors may consider further discussing the clinical relevance of their findings and potential “real world” application (e.g., timing of measurements, surveillance).

Author Response

Dear reviewer, thank you very much for all your comments. In regards with the first point, all patients underwent complete surgical resection. This has been clarified in the corrected manuscript.

Considering the second point, the type of adjuvant treatment administered to the patients has been included in Table 1 of the corrected manuscript.

Finally, a brief discussion of the clinical relevance and potential application of these findings has been included at the end of the discussion section of the corrected manuscript.

All changes in the corrected manuscript have been done with the “track changes” in yellow to allow easy visualization.